# Biofilm-Producing Ability of *Staphylococcus aureus* Obtained from Surfaces and Milk of Mastitic Cows

**DOI:** 10.3390/vetsci10060386

**Published:** 2023-06-02

**Authors:** Mária Vargová, František Zigo, Jana Výrostková, Zuzana Farkašová, Ibrahim F. Rehan

**Affiliations:** 1Department of the Environment, Veterinary Legislation and Economy, University of Veterinary Medicine and Pharmacy, Komenského 73, 04181 Košice, Slovakia; 2Department of Animal Nutrition and Husbandry, University of Veterinary Medicine and Pharmacy, Komenského 73, 04181 Košice, Slovakia; zuzana.farkasova@uvlf.sk; 3Department of Food Hygiene, Technology, and Safety, University of Veterinary Medicine and Pharmacy, Komenského 73, 04181 Košice, Slovakia; jana.vyrostkova@uvlf.sk; 4Department of Husbandry and Development of Animal Wealth, Faculty of Veterinary Medicine, Menoufia University, Shebin Alkom 32511, Egypt; ibrahim.rehan@vet.menofia.edu.eg; 5Department of Pathobiochemistry, Faculty of Pharmacy, Meijo University, Yagotoyama 150, Tempaku-Ku, Nagoya-Shi 468-8503, Japan

**Keywords:** mastitis, *Staphylococcus aureus*, biofilm, milking, parlor, environment

## Abstract

**Simple Summary:**

The ability of *S. aureus* to attach to different surfaces and assist with biofilm formation is described in many types of environments, but mainly in the milk processing environment. The development of biofilm is one of the most significant virulence mechanisms involved in the adherence of staphylococcal strains to living or non-living surfaces. Recently, many studies have revealed that biofilms formed in milking equipment may be a source of ongoing *S. aureus* contamination. The results of this study indicate that *S. aureus* was the most represented pathogen in milk and on surfaces. In the current study, the biofilm-producing ability of the reference strain and isolates of *S. aureus* obtained from surfaces and milk was determined. In all strains, the counts necessary for biofilm formation (>5 Log_10_ CFU/cm^2^) were detected, except for the reference strain. The ability to produce biofilm from isolates of *S. aureus* was higher in comparison with the reference strain during the first 3 h. In addition, significant differences between types of environmental contamination—occurrence of *S. aureus* on the floor, teat cup, and cow restraints, and the frequency of mastitis caused by *S. aureus* (*p* < 0.05)—were confirmed, which potentially poses a serious risk of this pathogen persisting in the milking environment and during biofilm formation.

**Abstract:**

This study was conducted to evaluate the incidence of mastitis in 153 dairy cows and to evaluate the kinetics of adhesion of isolates obtained from surfaces and milk in comparison with the reference strain (RS), CCM 4223. The surfaces of the floor, teat cup, and cow restraints were aseptically swabbed in three replicates (n = 27). Of the total number of infected cows (n = 43), 11 samples were found to be positive for *Staphylococcus aureus*, 12 samples tested positive for non-aureus staphylococci, 6 samples tested positive for *Streptococcus* spp., and 11 samples tested positive for other bacteria (*Escherichia coli, Pseudomonas* spp.) or a mixed infection. The most represented pathogen in milk (11/43) and on surfaces (14/27) was *S. aureus.* The kinetics of adhesion of the reference strain and isolates of *S. aureus* on stainless steel surfaces were determined after 3, 6, 9, 12, 24, and 48 h, and 3, 6, 9, 12, and 15 days of incubation. All strains reached counts higher than 5 Log_10_ CFU/cm^2^ needed for biofilm formation, except RS (4.40 Log_10_ CFU/cm^2^). The isolates of *S. aureus* revealed a higher capability to form biofilm in comparison with RS during the first 3 h (*p* < 0.001). Thus, there is a significant difference between the occurrence of *S. aureus* on monitored surfaces—floor, teat cup, and cow restraints—and the frequency with which mastitis is caused by *S. aureus* (*p* < 0.05). This finding raises the possibility that if various surfaces are contaminated by *S. aureus*, it can result in the formation of biofilm, which is a significant virulence factor.

## 1. Introduction

Mastitis is an inflammatory condition of the mammary gland that affects all dairy herds. It is a complex illness; its occurrence varies from herd to herd, and the number of ill cows is frequently unknown on most dairy farms. As a result, dairy cow herds need to constantly work on prevention and control [1,2]. Intramammary infections (IMI) involving bacteria are the main cause of mastitis [3]. Important mastitis pathogens have a variety of primary habitats due to their specific needs in terms of environmental variables [1,2].

Causal agents of IMI have been identified in more than 140 species of microorganisms. Up to 95% of instances involving infection are caused by pathogenic bacteria that enter the mammary gland through the teat canal [4]. The most prevalent mastitis-causing bacteria can be split into two categories: infectious pathogens and environmental pathogens. *Staphylococcus aureus*, *Streptococcus agalactiae*, and *Streptococcus dysgalactiae* are examples of contagious pathogens that can survive and grow inside the mammary gland, meaning that transmission from infected to uninfected quarters, and transmission from cow to cow, is most likely to occur during this time [5]. Environmental pathogens occur mainly in the environment. *Streptococcus uberis*, non-aureus staphylococci (NAS), and *E. coli* are the most significant members of this category, each having numerous strains with different levels of pathogenicity for both people and animals [2]. However, in accordance with Cobirka et al. [5], a number of microorganisms, including *Str. agalactiae* and *S. aureus*, can be categorized as environmental pathogens even though they are typically cited in the literature as being causes of contagious mastitis. This is because these pathogens can spread through multiple channels, including feces, urine, bedding, and other contaminants, in addition to contaminated milk from infected cows.

Staphylococci are ubiquitous in the dairy cow’s environment, and *S. aureus* is recognized worldwide as being a frequent cause of clinical or subclinical mastitis. Many sources of *S. aureus* have been identified in the milking parlor setting [6,7]. As the infected mammary gland serves as the main *S. aureus* reservoir, keeping the udder clean and milking regularly can shield a healthy cow from an infected cow, hence lowering the chances of infection [8]. *S. aureus* was discovered by Bogdanovičová et al. [7] in raw milk and milk processing tools (milk filters) in a study examining 50 dairy farms (from 2012 till 2014) in the Czech Republic. The author found *S. aureus* in 58 samples from 261 raw milk and milk filters, of which, 37 (14.2%) were isolated from raw milk and 21 (8.1%) were isolated from milk filters. When 42 dairy farms in the west of Slovakia were examined, Holko et al. [9] verified a significant incidence of IMI caused by NAS and *S. aureus*, which had been isolated from contaminated milk samples. NAS was the most frequently found bacterium, accounting for 35.9% of positive findings. *S. aureus*-related IMI were found in 12.5% of cases, mostly as a result of clinical mastitis.

The development of biofilm is one of the key virulence mechanisms involved in the adherence of Staphylococcal strains to living or non-living surfaces [10]. Several studies performed on dairy farms confirmed the ability of *S. aureus* isolates to produce biofilm [11,12]. *S. aureus* normally lives in environments where planktonic cells cling to different surfaces, multiply, and build into multi-layered cell clusters that are enclosed in an organic polymer matrix, or biofilm [13]. Compared with situations concerning vulnerable and exposed planktonic cells, this format protects the bacterial population from antibiotic attacks, environmental challenges, and the host’s immune system [14]. This could increase the likelihood of cross-contamination and economic losses due to *S. aureus’* persistence in various environments. The ability of *S. aureus* to attach to surfaces has been reported in the milk processing environment, mainly on surfaces of stainless steel or rubber [15]; these surfaces can be contaminated by different types of bacteria, which, under special conditions, adhere to those surfaces and cause biofilm formation [16]. *S. aureus* has demonstrated that it can adhere to surfaces such as stainless steel, polystyrene, glass, and polypropylene [17,18,19,20].

Biofilm formation and biofilm structure change dynamically depending on environmental conditions. Several environmental factors influence biofilm formation, including osmolarity (high osmolarity in the case of *S. aureus*), cell surface hydrophobicity, optimum temperature, and nutrient content. Some bacterial cells transfer from mature biofilm to a planktonic form; then, these dispersed cells attach to a new surface [21,22]. Active dispersal is triggered by changes in the environment, such as temperature change, nutrient starvation, oxygen shortage, and metabolite buildup [22].

In accordance with Melchior et al. [23], in their study focusing on staphylococci isolated from mastitis milk in cows, the most common features of virulence in staphylococci strains isolated from clinical mastitis were biofilm development and antibiotic resistance. Biofilm formation represents a virulence factor, especially for *S. aureus*, and processes such as growth, colonization, and maturation, that follow bacterial attachment, are essential for the dissemination and persistence of staphylococci [24,25]. In strains from clinical and recurring episodes of mastitis, following prior ineffective therapy, there was an increase in biofilm development. The emergence of antimicrobial resistance is a rapidly increasing problem worldwide [26]. There are several reasons for biofilm’s increased resistance to biocides. Biofilms are highly organized environments with physiological variability at a phase interface and they have spatial heterogeneity [27]. In the matrix, organisms can also exchange genetic material and leave behind enzymes, such as proteases and β-lactamases, that can hydrolyze β-lactams and some biocides. An essential aspect of biofilm control is quorum sensing, which involves cell-to-cell signaling [27,28]. Early stress responses, which entail the activation and expression of novel genes, may also play a role in the survival of biofilm cells exposed to biocides [29,30]. Moreover, the diffusion of biocides and possible reaction with biofilm constituents, must be considered. As they act as a barrier, biofilm-forming bacteria which are enclosed in a matrix acquire properties that render them highly tolerant to UV light, antibiotics, host immune responses, chemical biocides, and other stresses from the external environment [31,32,33,34,35] such as extreme pH and temperature, poor nutrients, and high salinity or high pressure [36]. Recently, a great deal of research has indicated that biofilms in milking equipment may be a potential source of lingering *S. aureus* contamination.

The hygiene of the environment and providing therapy to dairy cows are two of the most important components of a program to control mastitis. A hygienic environment reduces the occurrence of pathogens, and the main effect of therapy is that it increases the rate at which infections are eliminated. Both these practices significantly decrease bacterial spreading, transmission, and subsequent intramammary infection [37]. Everything that comes into contact with an infected udder is a potential source of pathogens. Reducing the transmission of pathogens, such as *S. aureus* and *Str. agalactiae*, from udder to udder, or from udder to another material—milkers’ hands, udder cloths, and teat cups—lead to a reduction in the incidence of intramammary infections. The time between milkings is also a period in which infections are transmitted. This is because infections can be transmitted as a result of contaminated bedding, licking of the udder and teats, and teat contact with the tail switch or the back legs.

Therefore, this study aimed to analyze the incidence of mastitis by determining the presence of individual pathogens, evaluating the kinetics of adhesion of isolates obtained from surfaces and milk in comparison to the reference strain, and determining the relationship between the occurrence of *S. aureus* on monitored surfaces—floor, teat cup, and cow restraints—and the frequency of mastitis caused by *S. aureus*.

## 2. Materials and Methods

### 2.1. Dairy Cows and Udder Examination

This study was conducted on a dairy farm, located in Eastern Slovakia, which practices conventional (non-organic) farming. A herd of 230 dairy cows—Slovak spotted cattle breeds between their first and fourth lactation cycle—was used. The free housing scheme for dairy cows included straw bedding, and they had unlimited access to water. To meet the nutritional needs of a 650 kg cow, all cows were fed a total mixed ration made up of silage, hay, and concentrate, in accordance with international regulations [38]. The annual milk yield (305 d) was 8.405 kg per energy-corrected milk/cow, with a bulk milk somatic cell count of 178 × 10^3^ cells/mL. The milk was transported once daily to a milk processing factory and then pasteurized. The cows were milked twice daily in a herringbone milking parlor (DeLaval, Sweden). The study involved 153 lactating cows (77 cows were dry or post-calving), and the udder health was assessed using the California mastitis test CMT (Indirect Diagnostic Test, Krause, Denmark), a clinical examination, a sensory analysis of the milk from udder quarter fortification, and veterinary history (Figure 1). In accordance with Tančin [39], the 2 mL reagent was mixed with 2 mL milk from each quarter on the plate, and the results were scored as negative, trace, or affirmative (score 1–4) based on the formation of gel in the sample. From all cows, 12 mL of composite milk were aseptically collected as samples for bacteriological cultivation, following the guidelines of the National Mastitis Council [40]. The samples were immediately taken to the lab, cooled to 4 °C, and examined the next day.

### 2.2. Sanitation of Milking Equipment

The following protocol was used for routine washes of the milking machine and milk liners: (1) pre-rinse cycle using water; (2) wash cycle using a cleaning regimen that included active ingredients such as AMP detergent, sodium hydroxide up to 10%, and sodium hypochlorite solution up to a maximum of 90%; and (3) an acid wash cycle using a clean-in-place (CIP) acid cleaner and active ingredients such as AMP acid, 10% hydrogen peroxide, and 45% trihydrogenphosphoric acid. The time and temperatures of the pipeline and bulk tank washes were monitored and recorded by Milk Guard (Dairy Cheq Inc., Ontario, Canada).

### 2.3. Microbiological Examination of Milk Samples

Collected milk samples were spread onto a blood agar with 5% of sheep blood (Oxoid Deutschland GmbH, Wesel, Germany), MacConkey agar (Oxoid, Hampshire, UK), and Sabouraud Dextrose Agar with chloramphenicol (Oxoid, Hampshire, UK), in accordance with the German Veterinary Association’s guidelines [41]. A total of 10 µL of each milk sample was incubated at 37 °C for 24 to 48 h. The plates were analyzed after a 24- and 48-h incubation period at 37 °C. Bacteria were identified based on colony morphology, Gram staining (Euromex, Holland), and biochemical tests. Samples were classified as positive if the growth of one or more contagious udder pathogens, such as *S. aureus, Str. dysgalactiae*, or *Str. agalactiae*, were discovered. Moreover, the sample was considered positive if the growth of one or two environmental species was confirmed, or if the growth occurred in conjunction with the growth of the contagious pathogen. If infectious pathogens did not develop, and three or more environmental pathogens were isolated from a single milk sample without a contagious pathogen, the grown samples were deemed to be contaminated.

The Gram-positive cocci grown on blood agar were distinguished into *Streptococcus* spp. (catalase negative) and *Staphylococcus* spp. (catalase-positive) by performing the catalase test. A coagulase test was performed to classify catalase-positive staphylococci. Catalase-positive staphylococci isolates showing positive coagulase, mannitol, DNase, and α- and β-hemolysis were determined as being isolates of *S. aureus*. Coagulase-negative staphylococci were identified in accordance with Holko et al. [9]. In accordance with studies by Ozbey et al. [42], all presumed *Staphylococcus* spp. from the milk cultures were verified using a matrix-assisted laser desorption/ionization (MALDI-TOF) biotyper (Bruker Daltonics, Leipzig, Germany). Standard *S. aureus* CCM 4750 and *S. chromogenes* CCM 3386 (Czech Collection of Microorganisms, Brno, Czech Republic) were utilized to check for good quality.

On the basis of tiny, transparent colonies on the blood agar and subcultures on Edward’s agar medium (Oxoid, Hampshire, UK), the streptococci were identified. Based on typical color, growth, and the morphological and hemolytic characteristics, suspected streptococci were identified via biochemical tests, including sodium hippurate, catalase, and esculin hydrolysis in accordance with El-Aziz et al. [43].

The presence of enterococci in a primary culture was confirmed via Gram-staining and sub-cultivation on MAC agar and SlaBa-plates agar (Slanetz and Bartley, Medium, Oxoid Ltd., Basingstoke, UK), noting the growth and color of characteristic colonies. *Streptococcus* spp. colonies and confirmed strains of the *Enterobacteriaceae* family were biochemically identified at the species level using the STREPTOtest 24 and ENTEROtest 24 (both from Erba Lachema, Brno, Czech Republic), and the software, TNW Pro 7.0 (also from Erba-Lachema, Brno, Czech Republic). The probability of correct species identification was above 90% (Table 1).

### 2.4. Mastitis Forms

The National Mastitis Council [40] gave a matching mastitis grade to each mastitis case, with the mastitis grade being divided into severity levels. During the CMT examination, subclinical mastitis (SM) was defined as a high somatic cell count (SCC), but no milk abnormalities or obvious symptoms of systemic or local inflammation. Clinical mastitis (CM) was classified in accordance with three degrees of severity, namely: mild mastitis (CM1), characterized by visible changes in secretion without signs of inflammation of the mammary gland; moderate mastitis (CM2), characterized by local signs of inflammation of the mammary gland with visible changes in milk secretion; and dairy cows with severe mastitis (CM3) also showed general disorders such as fever, low temperature, loss of appetite, or inability to stand.

### 2.5. Samples from Surfaces and Microbiological Examination

Samples from surfaces (n = 27) were taken from 3 cows’ stands. From each cow’s stand, 3 samples were obtained from the floor, teat cups, and dairy cow restraint (Table 2). These samples were collected 3 times: the first was collected after milking the first 51 dairy cows; the second was collected after milking the next 51 dairy cows; and the third was collected after milking the last 51 dairy cows. Samples from the same location were collected, transported to the lab, and processed. Using a sterile cotton swab, pre-wetted in a physiological solution, microbiological swabs from surfaces (5 × 2 cm^2^) were performed on three duplicates, taken from the same surfaces, to capture microorganisms. Swabs were taken before and after a cleaning regimen, which involved rotating the cotton swab when it was in contact with the surfaces being examined. The swabs were placed in a sterile container with 10 mL of a sterile saline solution, and they were vortexed for 2 min to dislodge the bacteria.

From all pathogens obtained from the evaluated surfaces and milk, kinetic adhesion was only assessed for strains of *S. aureus* found on surfaces (n = 14) and in milk (n = 11). The bacterial suspension of *S. aureus*, used as inoculum, was diluted with a sterile saline solution (0.85 g/100 mL), resulting in a final concentration of approximately 8 Log of colony forming units per ml (CFU/mL), which was adjusted in accordance with the turbidity of the 0.5 McFarland standard tube [44]. The initial suspension and decimal dilution were prepared in accordance with STN EN ISO 6887-5 [45].

Using the Baird–Parker selective arbitration medium, staphylococci isolates from the studied samples were obtained in accordance with ISO 6888-1 [46]. The injected plates underwent a 24-h incubation period at 37 °C. Plates containing more than 10 and fewer than 150 atypical and typical colonies, respectively, were then used for the staphylococcal counts. On the surface of Columbia blood agar (Oxoid Ltd., Basingstoke Hants, UK), two typical colonies (1.0–1.5 mm colony, black or gray colonies with halo) and two atypical colonies (black or gray colonies without halo) were inoculated with sterile bacterial loops and incubated at 37 °C for 24 to 48 h, depending on their distinctive appearance.

Using the selective diagnostic medium, Violet Red Bile Agar, the quantity of bacteria from the family *Enterobacteriaceae* were counted in accordance with STN EN ISO 21528-1 [47] (VRBL; HiMedia, India). Individual strains were used for MALDI-TOF MS identification following incubation. Identification using MALDI-TOF MS was performed with Flex Analysis software, version 3.0, on an Ultraflex III apparatus, and BioTyper software, version 1.1. (Bruker Daltonics, Billerica, MA, USA). For the most precise examination, the individual samples were made using an extraction technique involving ethanol and formic acid.

The reference strain of *S. aureus* CCM 4223 (Czech Collection of Microorganisms, Brno, Czech Republic) was used for a comparison with isolates to determine how well biofilms could form. The reference strain was stored in Petri dishes on a medium GSP (GSP agar, cat. n. 1.10230.0500, Merck KGaA, Germany, a selective agar for *S. aureus*), at a temperature below 4 °C. The reference strain used in our study was obtained from overnight cultures grown on GSP agar at 37 °C. Using MALDI-TOF, the most abundant isolate was identified as being *S. aureus*, which was subsequently used to determine the ability to form a biofilm.

### 2.6. Stainless Steel Coupons and Biofilm Preparation

Coupons with the dimensions, 20 × 20 × 10 mm, made of stainless steel (STN 17 240, 17 241 W Nr. 1.4301 AISI 304), were used for the experiments. The stainless steel coupons were modified in several steps: (1) cleaned and sanitized using pure acetone; (2) immersed in a neutral detergent for an exposure time of 1 h; (3) flushed with sterile distilled water; (4) fried and cleaned with alcohol (70% *v*/*v*); (5) washed with sterile water; (6) dried for 2 h at 60 °C; and (7) sterilized in an autoclave at a temperature of 121 °C for 15 min [47].

One hundred and thirty-two stainless steel coupons were immersed in a solution of 100 mL of Brain–Heart infusion (BHI) (HiMedia, India) and 10 mL of the aforementioned inoculum in Petri dishes. The coupons in Petri dishes were incubated at room temperature for 3, 6, 9, 12, 24, and 48 h, and after 3, 6, 9, 12, and 15 days. The coupons were mixed occasionally during the incubation period. Subsequently, after each exposure time, the coupons (four for each exposure time) were removed from the solution and washed with a sterile saline phosphate-buffered solution (PBS; pH = 7.4) for 15 s, thus enabling the release of non-adherent cells. Cells attached to the coupons were collected by rubbing their surfaces with moistened cotton swabs, and they were resuspended in sterile peptone water (SPW; 0.1 g/100 mL) and vortexed for 30 s. Subsequently, the mixture was serially diluted (10^−1^–10^−5^) in SPW, and an aliquot of 100 µL was spread onto the sterile Baird–Parker Agar (BPA) plates which were incubated for 24 hr at 37 °C [48,49]. After the incubation period, the number of viable cells was counted, and the results were expressed in Log_10_ CFU/cm^2^ (Figure 2, Figure 3 and Figure 4).

### 2.7. Data Analysis

Counts obtained for the biofilm formation of reference strains and isolates were converted to decimal Logarithmic values (Log_10_ CFU/cm^2^) and submitted for an Analysis of Variance (ANOVA). Data were analyzed using the software, Graph Pad Prism, and a probability value of *p* < 0.05 was accepted as indicating a significant difference. Differences in means between the reference strain and the isolates on the stainless steel surface after 3, 6, 9, 12, 24, and 48 h, as well as after 3, 6, 9, 12, and 15 days, were evaluated using One Way ANOVA. The Bonferroni test for each exposure time was used as the post hoc Test. The Chi-square test was used to evaluate the relationship between environmental contamination—occurrence of *S. aureus* on monitored surfaces (floor, teat cup, and cow restraints)—and the incidence of mastitis caused by *S. aureus* in dairy cows.

## 3. Results

Figure 1 depicts the state of the udder health of 153 lactating dairy cows in accordance with the California mastitis test. Of 612 quarter milk samples, 508 (83%) were negative for CMT. In 92 (15%) quarter milk samples, positive CMT results, with a score ranging from 1 to 4, were recorded. In 10 dairy cows, 12 atrophied quarters without milk production were found during the udder examination.

Table 1 shows the numbers and percentages of isolates categorized by mastitis. From a total of 43 infected cows, 11 samples (25.6%) were found to be positive for *S. aureus*, 12 samples (28%) were positive for non-aureus staphylococci (NAS; *S. chromogenes, S. haemolyticus, S. warneri,* and *S. xylosus*), 6 samples (14%) were positive for *Streptococcus* spp. (*Str. uberis, Str. agalactie*, *Str. faecalis*), and 11 samples (25.6%) were positive for other bacteria (*E. coli, Pseudomonas* spp.) or mixed infections. The most common pathogen found in clinical mastitis was *Staphylococcus aureus* (25.6%). 

A summary of sample sources and representations of pathogens from the evaluated surfaces is shown in Table 2. Samples were isolated from different surfaces—floors (n = 9), teat cups (n = 9), and dairy cow restraints (n = 9). Of the total number (n = 27) of samples, 14 of them (52%) were positive for *S. aureus.*

Table 3 compares the prevalence of positive samples for bacterial pathogens in milk with the prevalence of positive samples for bacterial pathogens in the milking parlor’s environment. The highest level of incidence was found for *S. aureus*, which was isolated in 25.6% of milk samples and 51.9% of environmental samples. NAS bacteria were only isolated in milk samples (27.9%), and they were not present in the environment. Of the other environmental bacteria, the most frequently isolated were *E. coli* bacteria from milk (16.2%) and the environment (25.9%).

By statistically comparing the number of positive samples isolated from milk with the number of isolated samples from the milking parlor’s environment, we confirmed the dependence on the level of significance (α = 0.05) in *S. aureus* and *E. coli*. The value of the test criterion G ranged from 0.078 to 4.99, and in these two evaluations, the critical value of χ2 = 3.841 was reached, which resulted in the decision that the null hypothesis of dependence in the characters was confirmed. Thus, it was concluded that the positivity of the samples obtained from milk and the environment, as a test criterion, was not accidental (Table 3).

The kinetics of adhesion of the reference strain, *S. aureus* CCM 4223, to stainless steel surfaces after 3, 6, 9, 12, 24, and 48 h and 3, 6, 9, 12, and 15 days of incubation is shown in Figure 2a,b. The highest number of adhered cells of the reference strain (6.65 ± 0.13 Log_10_ CFU/cm^2^) on stainless steel surfaces was found after 48 h, with an initial decrease in the number of adhered cells after 3 d of incubation (5.54 ± 0.08 Log_10_ CFU/cm^2^) (*p* < 0.0001); and the second highest number was found after 12 d of incubation, when the cell counts increased to 5.43 ± 0.06 Log_10_ CFU/cm^2^ (*p* < 0.0001).

The kinetics of adhesion of isolates, obtained from surfaces, and their ability to adhere to stainless steel surfaces after 3, 6, 9, 12, 24, and 48 h and 3, 6, 9, 12, and 15 days of incubation, is shown in Figure 3a,b.

The number of adhered cells of isolates, obtained from surfaces, presented different phases wherein the number of adhered cells both increased and decreased; first, the number increased, with the number of adhered cells ranging from 5.20 ± 0.16 Log_10_ CFU/cm^2^ to 5.50 ± 0.22 Log_10_ CFU/cm^2^ after 3 and 6 h of incubation (*p* < 0.0001), followed by a decrease in the number of adhered cells from 5.50 ± 0.22 to 4.15 ± 0.13 Log_10_ CFU/cm^2^ after 12 h (*p* < 0.0001).

The kinetics of adhesion of isolates, obtained from infected milk, and their ability to adhere to stainless steel surfaces after 3, 6, 9, 12, 24, and 48 h and 3, 6, 9, 12, and 15 days of incubation, is shown in Figure 4a,b. The highest number of adhered cells of isolates from milk (7.85 ± 0.26 Log_10_ CFU/cm^2^) on stainless steel surfaces was found after 48 h of incubation (*p* < 0.0001). The number of adhered cells of isolates from milk increased from 5.84 ± 0.12 (after 3 h) to 7.85 ± 0.26 (after 48 h of incubation) Log_10_ CFU/cm^2^ (*p* < 0.0001); then, it decreased from 7.27 ± 0.27 to 3.92 ± 0.27 Log_10_ CFU/cm^2^ after 3 d and 15 d (*p* < 0.0001).

## 4. Discussion

In practical conditions on dairy farms, subclinical and clinical forms of mastitis are varied in terms of their tendency to comply with the milking program and hygienic conditions [4]. In our study, we detected the occurrence of mild and moderate clinical forms of mastitis at a level of 16.3%; these instances were mainly identified by a visual change in the milk secretion, which was detected during the forestripping of the milk. In 11.1% of cases, subclinical forms of mastitis were detected (Table 1), which often escape the attention of the farmer. An evaluation of the CMT and SCC in the milk is a procedure that is frequently utilized in practical settings [50]. By investigating 153 dairy cows, 15% of positive quarter milk samples was found to have a CMT score ranging from 1 to 4 (Figure 1). Based on the positive CMT and laboratory analyses, udder pathogens in 43 dairy cows were detected.

In many countries, mastitis is mainly caused by *S. aureus*, NAS, and streptococci [9,51]. The high prevalence of *S. aureus*-caused mastitis on dairy farms confirms that the pathogen holds the greatest level of pathogenicity due to the virulence factors that are required for the appearance of clinical signs. *S. aureus* is a ubiquitous microorganism in the milking parlor environment [52]. In our study, from a total of 43 infected cows, 11 milk samples tested positive for *S. aureus*, which was the most common pathogen in the studied cows (Table 1). Moreover, the most frequently isolated bacterium from surfaces was also *S. aureus* (Table 2). Any item in (or on) which *S. aureus*, an infectious agent, can be found, is a possible source of intramammary infection, and prominent reservoirs of *S. aureus* can be found in the infected mammary glands of milking cows. Many mastitis pathogens, such as *S. aureus*, *Str. agalactiae*, and *Str. uberis*, have been classified as contagious or environmental pathogens by Cobirka et al. [5]; this is because they can spread through a variety of channels, such as bedding, urine, feces, and other contaminants, in addition to contaminated milk from infected cows, or poor hygiene conditions during milking. Contagious mastitis pathogens such as *Str. dysgalactiae* and *Str. agalactiae* have been successfully controlled through major regulating programs that are based on antibiotic therapies, teat disinfection, and the culling of chronically infected cows [53], but intramammary infections caused by *S. aureus* and NAS continue to be a significant issue in dairy herds.

Currently, there are many studies focused on biofilms in food processing plants, but little is known about the presence of biofilms on dairy farms. *S. aureus* strains have been found in milking parlor environments, and they have been shown to be capable of forming biofilms in the past [54,55]. According to Lee et al. [11], *S. aureus* strains isolated from dairy farms are able to form biofilms on stainless steel, polystyrene, and rubber, but not silicone surfaces. By comparing the dependence of positive bacterial isolates from milk with bacterial isolates from the environment (floor, teat cup, and cow restraints), we confirmed dependence using the level of significance (*p* < 0.05) in *S. aureus* (Table 3). Our data confirm the statement that milk from dairy cows infected by *S. aureus* may be a continuous source of *S. aureus* contamination for the surfaces of milking equipment and other surfaces. The subsequent colonization of surfaces may lead to the eventual formation of biofilms [56]. The various microbial communities found in the biofilms generated on or in milking equipment may have included *S. aureus* [57].

Numerous variables, including species, culture medium type, temperature, and microbe concentration, might influence the adhesion process [58]. In accordance with Morton et al. [59], the adhesion process intensifies to its maximum point when microorganisms are allowed to grow at their optimum temperature, regardless of the species or surface employed. Pagedar et al. [60] detected a higher cell count of *S. aureus* at 25 °C than at 37 °C after 48 h; these biofilms were shaped on stainless steel surfaces. In our investigation, we incubated the samples at room temperature (23–25 °C), which is not the best environment for the growth of *S. aureus*. Despite this, the highest numbers of adhered cells of the reference strain were already found after 48 h of incubation. Although certain bacteria demonstrated a significantly stronger biofilm formation over the first 48 h at 25 °C, most isolates noted in the study by Vázquez-Sánchez et al. [61] exhibited higher biofilm production at 37 °C.

In accordance with the results shown by the kinetics of isolates from milk, the highest number of adhered cells of isolates from milk on stainless steel surfaces was found after 48 h of incubation. The production of biofilm requires a minimum of 5–6 Log_10_ CFU/cm^2^, whereas lower counts might only represent the adhesion process [62]. These findings conflict with those of Zotolla and Sasahara [63], who stated that lower counts would only indicate the adhesion process and higher counts (7–8 Log_10_ CFU/cm^2^) would be necessary for biofilm development.

Data observed in the present study showed that the cell count needed for biofilm formation in the reference strain was formed after 6 h of incubation, and the cell count needed for biofilm formation in the isolate strain was formed after 3 h of incubation. In accordance with the results presented by Marques et al. [48], after 3 h of contact, the count of attached cells of reference strain was 1.2 Log_10_ CFU/cm^2^, and after six hours, it was 4.27 Log_10_ CFU/cm^2^. These authors reported that the number of adhered cells correlates with the contact time, and the number of adhered cells would likely increase if a longer contact time had been used. Data observed in our study confirm these results; we detected an increase in the number of cells in the reference strain, from 4.40 Log_10_ CFU/cm^2^ of adhered cells after 3 h of contact to 5.05 Log_10_ CFU/cm^2^ of adhered cells after 6 h. On the other hand, the results of our study are not in accordance with the findings of Marques et al. [48], who detected bacterial counts of 7 and 8 Log_10_ CFU/cm^2^ after 15 days of cultivation; indeed, we found 5.57 Log_10_ CFU/cm^2^ of adhered cells after 15 days of incubation. Based on our results, the reference strain revealed a diminished ability to form biofilm in comparison with isolates of *S. aureus*; however, further research on the development of biofilm using the reference strain *S. aureus* is required because biofilms are a bacterial survival tactic that make them extremely difficult to treat due to their innate immune response and resistance to antibiotics and biocides.

The evaluated isolates revealed numbers of adhered cells between 4.15 and 5.40 Log_10_ CFU/cm^2^; these numbers of adhered cells are similar, unlike those detected by Meira et al. [64]. Their study found approximately 4.5 Log_10_ CFU/cm^2^ of adhered cells, and they also detected an initial decrease in several adhered cells of isolates after 48 h of incubation, followed by an increase after 72 h. Our findings are consistent with earlier research by Souza et al. [65], who discovered the largest concentrations of adherent cells (5–6 Log_10_ CFU/cm^2^) on stainless steel surfaces over a 72-h period.

## 5. Conclusions

The present study examines the most represented pathogen in milk, *S. aureus* (25.6%), followed by non-aureus staphylococci (*S. chromogenes, S. haemolyticus, S. warneri, S. xylosus*), *Streptococcus* spp. (*Str. uberis, Str. faecalis*), and other bacteria (*E. coli, Pseudomonas* spp.). Moreover, the most represented pathogen on surfaces was *S. aureus* (52%). According to the kinetics of adhesion of the reference strain and the isolates of *S. aureus* and their ability to adhere to stainless steel surfaces, all strains (except the reference strain (4.40 Log_10_ CFU/cm^2^)) reached counts higher than 5 Log_10_ CFU/cm^2^, which is needed for biofilm formation. The isolates of *S. aureus* revealed a greater ability to form a biofilm in comparison with the reference strain during the first 3 hr (*p* < 0.001). Based on our results, almost 26% of pathogens isolated from milk, and 52% of pathogens isolated from surfaces in the milking parlor environment, were *S. aureus*, which possesses the ability to produce biofilms on stainless steel coupons, thus indicating the persistence of this pathogen in the milking environment.

We found a statistically significant relationship between the occurrence of *S. aureus* on monitored surfaces—floor, teat cup, and cow restraints—and the incidence of mastitis caused by *S. aureus* (*p* < 0.05); this suggests that the presence of *S. aureus* on different surfaces can lead to biofilm formation, which is an important virulence factor, and it can act as a source of *S. aureus* contamination for dairy cows and on other surfaces. The relationship between *Staphylococcus aureus* and mastitis cases, as well as their occurrence on surfaces and in milk, emphasizes the need to improve hygiene practices to prevent biofilm formation on dairy farms.

## Figures and Tables

**Figure 1 vetsci-10-00386-f001:**
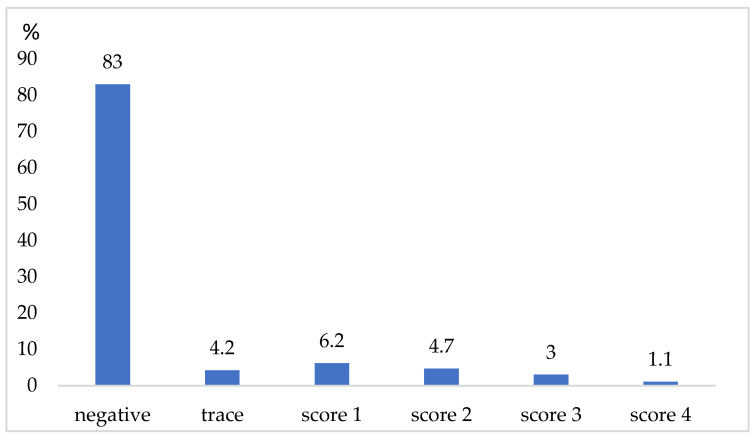
California mastitis test scores from 612 tested mammary gland quarters (CMT—Californian Mastitis Test with the scores: negative; trace; score 1, 2, 3, and 4).

**Figure 2 vetsci-10-00386-f002:**
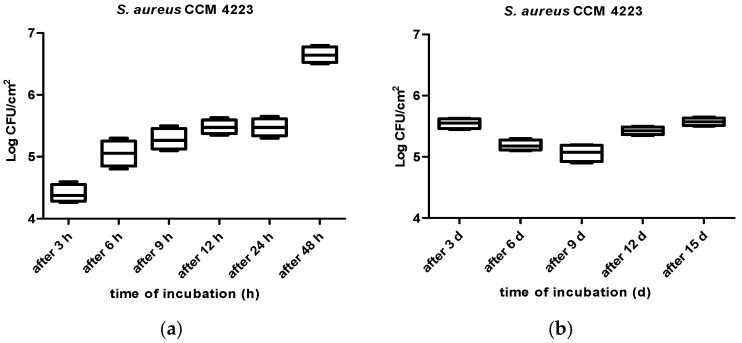
(**a**,**b**) Kinetics of adhesion of the reference strain, *S. aureus* CCM 4223, and their ability to adhere to stainless steel surfaces after 3, 6, 9, 12, 24, and 48 h and 3, 6, 9, 12, and 15 days of incubation.

**Figure 3 vetsci-10-00386-f003:**
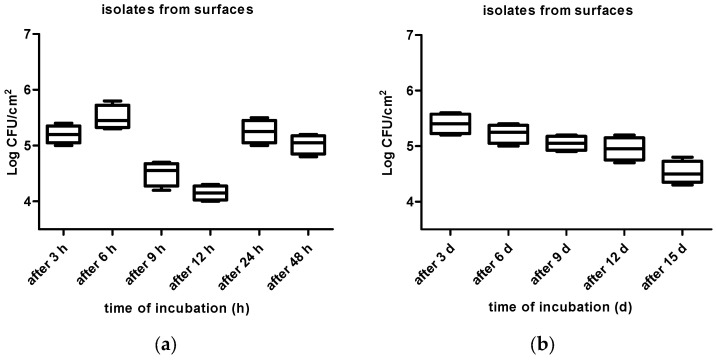
(**a**,**b**) Kinetics of adhesion of isolates, obtained from surfaces, and their ability to adhere to stainless steel surfaces after 3, 6, 9, 12, 24, and 48 h and 3, 6, 9, 12, and 15 days of incubation.

**Figure 4 vetsci-10-00386-f004:**
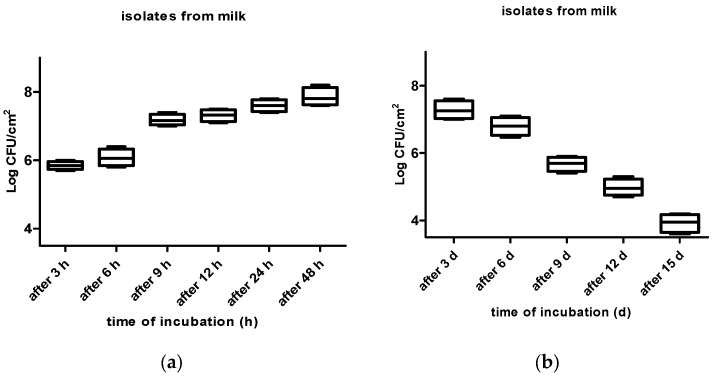
(**a**,**b**) Kinetics of adhesion of isolates obtained from milk, and their ability to adhere to stainless steel surfaces after 3, 6, 9, 12, 24, and 48 h and 3, 6, 9, 12, and 15 days of incubation.

**Table 1 vetsci-10-00386-t001:** Isolated microorganisms from mastitic cows in the monitored herd.

Isolated Bacteria			Subclinical ^2^		Clinical ^3^					
					CM1		CM2		CM3	
	n	%	n	%	n	%	n	%	n	%
*Staphylococcus* spp.										
*S. aureus*	11	25.6	2	4.7	4	9.3	4	9.3	1	2.3
*S. intermedius*	3	7.0	--	--	3	7.0	--	--	--	--
NAS ^1^										
*S. chromogenes*	5	11.6	3	7.0	--	--	2	4.7	--	--
*S. haemolyticus*	3	7.0	--	--	2	4.7	1	2.3	--	--
*S. warneri*	2	4.7	1	2.3	1	2.3	--	--	--	--
*S. xylosus*	2	4.7	2	4.7	--	--	--	--	--	--
*Streptococcus* spp.										
*Str. uberis*	4	9.3	1	2.3	2	4.7	1	2.3	--	--
*Str. faecalis*	2	4.7	2	4.7	--	--	--	--	--	--
Other bacteria										
*E. coli*	4	9.3	2	4.7	--	--	2	4.7	--	--
*Pseudomonas* spp.	3	7.0	3	7.0	--	--	--	--	--	--
*Mixed infection **	4	9.3	1	2.3	2	4.7	--	--	--	--
Total	43	100	17	39.5	14	32.5	10	23.2	1	2.3

Note: NAS ^1^—non-aureus staphylococci; n—number of isolated bacteria from examined samples. Mixed infection *—mixed infection caused by two or more bacterial types; subclinical mastitis ^2^ is characteristic with a positive CMT score, bacteriological cultivation, increased SCC, and reduced milk yield without clinical signs; clinical mastitis ^3^ was classified as mild mastitis (CM1), moderate mastitis (CM2), and severe mastitis (CM3).

**Table 2 vetsci-10-00386-t002:** Summary of sample sources and representations of pathogens from the evaluated surfaces.

Sample Source	n	Isolated Bacteria	Positive for *S. aureus* n (%)
Floor	9	*Streptococcus* spp. *S. aureus* *E. coli* *Enterococcus faecalis*	(15%)
Teat cup	9	*S. aureus* *E. coli* *Campylobacter* spp.	7 (26%)
Dairy cow restraints	9	*S. aureus* *Hafnia alvei* *Ralstonia insidiosa* *E. coli*	3 (11%)
Total	27		14/(52%)

**Table 3 vetsci-10-00386-t003:** The comparison of the dependence of positive bacterial isolates from milk with isolates from the environment (floor, teat cup, and cow restraints).

Isolated Bacteria	Sample Source								Testing Value	*p*-Value
	Floor (n)	%	Teat Cup (n)	%	Cow Restraints (n)	%	Milk (n)	%	χ^2^	*p*
*S. aureus*	4	15	7	26.0	3	11.1	11	25.6	4.99 *	0.025 *
NAS	0	0	0	0	0	0	12	27.9	0.078	0.780
*Streptococcus* spp.	1	3.7	0	0	0	0	6	14.0	1.93	0.164
*E. coli*	2	7.4	1	3.7	4	14.8	7	16.2	4.13 *	0.042 *
Other bacteria	2	7.4	1	3.7	2	7.4	7	16.2	0.95	0.329
Total	9	33.5	9	33.2	9	33.3	43	100		

Note: NAS—non aureus staphylococci represent *S. warneri*, *S. chromogenes*, and *S. xylosus*; * Chi-squared test at a significance level α = 0.05; critical value χ^2^ = 3.841; a positive testing value (G) regarding the statistical dependence of bacteria isolated from milk and the environment was confirmed when G > χ^2^; the dependence was not statistically significant when the testing value was G < χ^2^.

## Data Availability

All existing data are listed in the manuscript.

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
