# Peer review of "Biofilm-Producing Ability of Staphylococcus aureus Obtained from Surfaces and Milk of Mastitic Cows"

_vetsci, 2023, doi:10.3390/vetsci10060386_

Round 1

Reviewer 1 Report

This research investigated the presence of S. aureus biofilm formation on various surfaces of a commercial dairy located in Eastern Slovakia.  The study design was appropriate; however, there are many instances of grammatical errors and poor sentence structure, which makes reading and comprehending the manuscript very difficult.  I would suggest a major revisions before resubmitting. 

Introduction:

The Introduction is largely a general discussion about mastitis and pathogenic and nonpathogenic (environmental) bacteria. More emphasis should be placed on S. aureus and biofilm formation.  Specifically, what environmental conditions encourage biofilm formation? How do biofilms contribute to antimicrobial resistance and to resistance to disinfectants used to clean the milking equipment? Is there a genetic basis to biofilm formation, if so, what are the mechanisms.  The introduction should also include previous research that is similar to this study.  I suggest lines 327-354 found in the Discussion should be condensed and moved to the Introduction.  

- Zoohygienic seems to be a commonly used word in papers in this research area, but I can’t find any definition of it. Lines 50-52 state that zoohygienic provisions are a health risk to humans and animals, but I am uncertain as to why this is so. 

- Line 54 - what non-infectious factors lead to mastitis?

- Line 58-61 - needs to be referenced.

- Lines 61-65 and 90-92 - both instances I am unsure what the authors are saying. 

Materials and Methods:

Just some minor editing is required for this section.

- Line 114 'realized' is not correct, use 'conducted'

- Line 149-151.  I'm uncertain why samples with more than one isolate were considered 'contaminated'. Why can't a milk sample be co-infected with more than one type of bacteria? Perhaps the 'contamination' is not the correct word?

- Line 201 - M&M should not be referencing Figures in the Results section. The data analysis section should be were the averaging of the kinetic values is discussed. 

- Rewrite line 260.  It should be very clear what parameters were tested in the Chi-square analysis -- "environmental contamination" is not an outcome parameter. Also, check the definition of prevalence and incidence to ensure you are using incidence correctly. 

Results:

Too much of the results section is describing what is already clearly displayed in the Figures and Tables.

There is no p-values in the tables/figures that relate back to the Chi-square analysis.

The description of the kinetics is tedious reading. It is clear from the Figures that the average Log values are changing. I would also suggest that Figures 2 and 3 should be labelled Figure 2a and 2b and so on with the remaining figures that have multiple panels.  You could also annotate to the graphs to show where statistical differences arise. 

Discussion:

As noted, the first three paragraphs of the discussion relate to background material, and there is no reference to the study results until until line 354 ("According to the kinetics...).  The discussion should start with the main findings and then discuss these in context with previous publications, or research conducted in dairies with other bacteria that form biofilms. 

Lines 371-373 - what constitutes a biofilm (CFU count) seems central to the paper, and you note that your definition was different than Zotella and Sasahara. This needs to be expanded on - is anyone using 5-6 Log 10 counts as you have done?

You should discuss the significance of the reference strain having lower CFU counts.

Why do the CFU counts increase and decrease - is this nutritionally related or is because the biofilm matrix has been established and there is a negative feedback loop on growth?

Can you explain the following, "No matter the species or surface used, the adhesion process intensifies to its fullest, according to Morton et al. [41], when microorganisms are let to grow at their optimum temperature. In our investigation, we incubated the samples at room temperature (23–25 °C), which is not the best environment for the growth of S. aureus."  Specifically, what is the optimal growth temperature - 37C?  And, what is the temperature of most of the surfaces in the barn - I assume room temperature?  It seems reasonable that you would culture at RT vs 37C since this represents what occurring in the barn. 

What were the limitations of the study?  The obvious one is that only one dairy was sampled. Perhaps S. aureus at other dairies would behave differently. 

The Simple Summary and Abstract both mention "a significant difference between environmental contamination and the frequency of mastitis brought on by S. aureus (p = 0.03)", but in the Conclusions you state "We found a statistically significant relationship between environmental contamination and the incidence of mastitis caused by S. aureus (p < 0.05)..." What is the correct p-value? 

Other comments

- Don't use slang such as "Nowadays" and "These days" 

- Avoid contractions such as "aren't" and "isn't"

- Be consistent in S. aureus in line 368 vs. Staph. aureus as in line 405

- In multiple places you state "These authors reported.. " or the "the authors discovered..."  There is no need to include authors.  Just state the finding and reference it appropriately, and no need to put the authors name and the citation in the same sentence. 

The manuscript should be revised by someone with a very strong command of the English language. 

Author Response

Dear Reviewers,

We have attached the revised manuscript for the research paper entitled “Biofilm-producing ability of Staphylococcus aureus obtained from surfaces and milk from mastitic cows.” We have responded to each of the comments and suggestions and provided explanations and descriptions of all changes made to the manuscript. First of all, we would like to thank the reviewers for their helpful comments and valuable questions. Please find our comments and description of the changes below. All changes made to the manuscript have been highlighted in yellow color. Note, if the article is accepted after your and the editor's decision, then final English editing will be done at the MDPI service.

As a result of these changes, the manuscript improved considerably – Thank you!

Best regards, Authors.

Author Response

(The authors gave the same response as above.)

Reviewer 3 Report

The manuscript describes isolation and identification of mastitis pathogens from composite milk samples from all lactating cows in a single farm as well as the isolation of staphylococci from cow environment. Furthermore it describes an ability of identified Staphylococci for biofilm formation and compare it with the ability of reference strain.

In the introduction part of the manuscript authors briefly described epidemiology of pathogens pointing out the contagious nature of staphylococcal pathogens. Ability of staphyloccoci for biofilm formation is offered as a rationale for high prevalence of staphyloccocal mastitis.

The aim of the research is clearly stated.

In the material and method section authors described applied methodology to examine udder health status, to take milk samples, to isolate and identify mastitis pathogens, to examine ability of isolated staphylococci for biofilm formation.

The sample size (cow level and the isolated staphylococcal strain level) does not require additional explanation since all (cows and isolates) were included in the study.

Authors described the applied methods, including statistical treatment, with sufficient details.

Minor request is directed to the description of microbiological examination of samples taken from the surfaces. Authors are asked to state whether these samples were examined using the same protocols as the milk samples or using a different protocol. If other protocols were used, authors are asked to offer additional explanation. Furthermore, if the same protocol was used to isolate and identify bacterial strains authors are asked to describe the decision making if more than two different genera had grown on nutrient media.

The statement (line 260/262): „The Chi-square test was used to evaluate the relationship between environmental contamination by S. aureus and the incidence of mastitis caused by S. aureus in dairy cows.” is unclear since is unsupported with the result in the Result section. The design of the research is cross-sectional so it is unclear how authors evaluated incidence.

Furthermore there is no statistical comparison between abundance of staphylococci of different origin at the same time points. Authors compared kinetics of the same strain, comparing abundance of staphylococci between different time points of the same strain. However they did not compare different strains at same time points.

Results are presented as self-explanatory tables and figures (graphs). A minor objection is directed toward graphical presentation of results. The presentation would be more convincing and logical if presented as a single unbroken graph for whole period (from 3 h to 15th day) or at least if two graphs relating the same origin of strain have the same scaling of y axes.

In the discussion section, authors compare the own results with the results of similar researches carried out elsewhere giving a reasonable explanation for found differences.

The conclusions are in line with the achieved results except for the incidence as already stated above in the comments of Material and method section.

Author Response

(The authors gave the same response as above.)

Reviewer 4 Report

Introduction.

The first two paragraphs should be shortened significantly, as they contain a lot of well-known information.

Procedures.

There are some mistakes in the terminology used by the authors (e.g., milk lines instead of milk liners, primo-cultivation instead of primary culture) and these should be corrected.

Results.

Figures are numbered badly, they need to be corrected.

Also, figures must be colorized.

Discussion.

This is shallow and really not interesting for readers. It must be extended and also authors must go into greater detail.

Some of the conclusions are not conclusions but really results and must be transferred accordingly.

Extensive editing of English language required

Author Response

(The authors gave the same response as above.)

Reviewer 5 Report

I had the opportunity to review the manuscript entitled “Biofilm-producing ability of Staphylococcus aureus obtained from surfaces and milk from mastitic cows”.   Many papers and review, suggest the presence of biofilms in milking facilities as a possible source of persistent S. aureus contamination. the authors provide the results of monitoring a dairy cow herd, indicating that the pathogen most represented in milk (25.6%) and surfaces (52%) is S. aureus. The authors studied the adhesion kinetics of the reference strain (RS CCM 4223) and S. aureus isolates to stainless steel surfaces, determined after 3, 6, 9, 12, 24, 48 hours and 3, 6, 9, 12, 15 days of incubation. All strains tested achieved counts above 5 Log10 CFU/cm2 required for biofilm formation, except RS (4.40 Log10 CFU/cm2). S. aureus isolates revealed a greater ability to form biofilm than RS during the first 3 hours (p < 0.001). In this case, a significant difference was found between environmental contamination and the frequency of mastitis caused by S. aureus (p = 0.03). This result raises the possibility that contamination by S. aureus of various surfaces may lead to the formation of biofilm, which is an important and non-negligible virulence factor.

Below my minor considerations line-by-line  

L. 58-59: “Staphylococcus aureus, Streptococcus agalactiae, and Streptococcus dysgalactiae are examples of contagious pathogens that can survive and grow inside”. Are the authors sure that Streptococcus dysgalactiae always behaves as a contagious pathogen?

L. 94:  “[10].Growth,”  check.

L 120: “with a bulk milk somatic cell count of 178 x 103 cells/mL. * 100”.” It is a value of geometric mean? Specify.

Line 122: “The cows were milked twice daily in a herringbone milking parlor (DeLaval, Sweden).” Add more informations eg vacuum level, cluster detachment (manual or automatic), milking routine …..

Line 126: (Figure 1). Tanin [17] states that the reagent was”. Check the author.

Line 132-140: 2.2. Sanitation of milking equipment. In this section the authors report numerical values, e.g. 866500 mg.kg-1 , but they should be checked again.

Line 203-204: “to a final concentration of approximately 8 Log of colony form- 203 ing units per mL (CFU/mL) which was adjusted according to the turbidity of 0.5 McFar- 204 land standard tubes (inoculum) [22].”. I suggest the authors provide a better explanation of this sentence.

Line 256-258: “Differences in an arithmetic mean between the reference strain and the isolates on the stainless steel surface after 3, 6, 9, 12, 24, and 48 hours as well as after 3, 6, 9, 12, and 15 days were evaluated  by One Way ANOVA.”. Check whether 'arithmetic mean' is the correct term?

Line 265: “Of of 612 quarter.”. Check “of”

Line 269: “Figure 1. Evaluation of CMT in 153 exanimated cows (CMT - Californian Mastitis Test with the score: negative; trace; score 1, 2, 3, and 4).”. Recheck sum of CMT values (%) on histograms and in the text.

Line 271-277: “Table 1 shows the numbers and percentages of isolates categorized by mastitis form.From a total of 43 infected cows, 11 samples (25.6%) were found positive for S. aureus, 12 samples (28%) for non-aureus staphylococci (NAS; S. chromogenes, S. haemolyticus, S. warneri, and S. xylosus), 6 samples (14%) for Streptococcus spp. (Str. uberis, Str. agalactie, Str. faecalis), and 11 samples (25.6%) for other bacteria (E. coli, Pseudomonas spp.) or mixed infection. The most common pathogen in clinical mastitis was Staphylococcus aureus (25.6%).”. The nomenclature of isolated bacteria must be written in italics; also check “form.From.

Line 309: “Figure 2,3. Kinetics of adhesion of reference strain S. aureus CCM 4223 to stainless steel surfaces after 3, 6, 9, 12, 24, 48 hours; and 3, 6, 9, 12, 15 days of incubation”. Unify the nomenclature of S. aureus in the two charts.

Line 353: “Nowadays, there are many studies focused on biofilms in food processing plants, 353 but little is known about the presence of biofilms on dairy farms”. Add a reference.

Line 361: [36,37,38]. Rewrite in correct form.

Line 362: “S. aureus [39].The”. Check.

Line 406-413: “We found a statistically significant relationship between environmental contamination and the incidence of mastitis caused by S. aureus (p < 0.05), which suggest that the presence of S. aureus on different surfaces can lead to biofilm formation as an important  virulence factor and it can act as a source of S. aureus contamination for dairy cows and surfaces. The relationship between Staphylococcus aureus and mastitis cases and their occurrence on surfaces and in milk emphasizes the need to improve hygiene practices to prevent biofilm formation on dairy farms”. These two sentences need to be revised and rewritten.

References: check references n. 3 “Zigo, F.; Vasil’, M.; Ondrašovičová, S.; Výrostková, J.; Bujok, J.; Pecka-Kielb, E. Maintaining Optimal Mammary Gland Health 435 and Prevention of Mastitis. Front. Vet. Sci. 2021, 8, 607311.”. I could not find it in the text.

I consider moderate editing of the English language

Author Response

(The authors gave the same response as above.)

Round 2

Reviewer 1 Report

This version is much improved. There are still a large number of grammatical and sentence structures issues. I have attempted to address many of them in my attached notes. 

You need to standardize the use of all abbreviations, including S. aureus. 

The statistical section is still a bit confusing as to what you analyzed, and you may want to consider using a Fisher's exact test. 

I found the Discussion a bit unstructured. I think you have interesting results, but the Discussion did not highlight what was unique about your study or the findings.  The message was that you found biofilms in the environment and they can be a source for mastitis and vice versa. I'm uncertain as to what other message I should have received. 

As above. 

Author Response

Dear Reviewer,

We have attached the corrected manuscript for the research paper entitled “Biofilm-producing ability of Staphylococcus aureus obtained from surfaces and milk from mastitic cows” after second revision. We have responded to each of the comments and suggestions and provided explanations and descriptions of all changes made to the manuscript. Please find our comments and description of the changes below. All changes made to the manuscript have been highlighted in yellow color. The manuscript was edited by MDPI English editing service with certification.

Reviewer 2 Report

Accepted after revision.

Author Response

(The authors gave the same response as above.)

Reviewer 4 Report

The manuscript has been improved.
However, there is still room for improvement of the discussion, especially by adding some further relevant references, which have not been included, and by indicating the clinical consequences of the study.

Moderate editing of English language.

Author Response

Dear Reviewer,

We have attached the corrected manuscript for the research paper entitled “Biofilm-producing ability of Staphylococcus aureus obtained from surfaces and milk from mastitic cows” after second revision. We have responded to each of the comments and suggestions and provided explanations and descriptions of all changes made to the manuscript. Please find our comments and description of the changes below. All changes made to the manuscript have been highlighted in yellow color. The manuscript was edited by MDPI English editing service with certification.

Response to Reviewer Comments:

Reviewer: However, there is still room for improvement of the discussion, especially by adding some further relevant references, which have not been included, and by indicating the clinical consequences of the study.

Response: Although it is accepted that biofilms are omnipresent in nature, the significance of biofilms from the point of clinical consequences, especially with regard to their role in  infections, is often underestimated. Biofilms represent the major challenge to microbiologists and specialists, and the biggest challenge stays elucidating the factors that make the biofilm phenotype different from the planktonic phenotype. From the perspective of the relationship between occurence of biofilm and mastitis, bacterial biofilms are a problem to be recognized and managed as they have definite infection consequences.
